# Expression Profiles and Characteristics of Apple lncRNAs in Roots, Phloem, Leaves, Flowers, and Fruit

**DOI:** 10.3390/ijms23115931

**Published:** 2022-05-25

**Authors:** Dajiang Wang, Yuan Gao, Simiao Sun, Lianwen Li, Kun Wang

**Affiliations:** Research Institute of Pomology, Chinese Academy of Agricultural Sciences (CAAS), Key Laboratory of Horticulture Crops Germplasm Resources Utilization, Ministry of Agriculture and Rural Affairs of the People’s Republic of China, Xingcheng 125100, China; dajiang0101@126.com (D.W.); gaoyuan02@caas.cn (Y.G.); sunsimiao@caas.cn (S.S.); lilianwen@caas.cn (L.L.)

**Keywords:** apple, lncRNA, tissue–specific expression, regulatory networks, qRT-PCR

## Abstract

LncRNAs impart crucial effects on various biological processes, including biotic stress responses, abiotic stress responses, fertility and development. The apple tree is one of the four major fruit trees in the world. However, lncRNAs’s roles in different tissues of apple are unknown. We identified the lncRNAs in five tissues of apples including the roots, phloem, leaves, flowers, and fruit, and predicted the intricate regulatory networks. A total of 9440 lncRNAs were obtained. LncRNA target prediction revealed 10,628 potential lncRNA–messenger RNA (mRNA) pairs, 9410 pairs functioning in a cis-acting fashion, and 1218 acting in a trans-acting fashion. Functional enrichment analysis showed that the targets were significantly enriched in molecular functions related to photosynthesis-antenna proteins, single-organism metabolic process and glutathione metabolism. Additionally, a total of 88 lncRNAs have various functions related to microRNAs (miRNAs) as miRNA precursors. Interactions between lncRNAs and miRNAs were predicted, 1341 possible interrelations between 187 mdm-miRNAs and 174 lncRNAs (1.84%) were identified. MSTRG.121644.5, MSTRG.121644.8, MSTRG.2929.2, MSTRG.3953.2, MSTRG.63448.2, MSTRG.9870.2, and MSTRG.9870.3 could participate in the functions in roots as competing endogenous RNAs (ceRNAs). MSTRG.11457.2, MSTRG.138614.2, and MSTRG.60895.2 could adopt special functions in the fruit by working with miRNAs. A further analysis showed that different tissues formed special lncRNA–miRNA–mRNA networks. MSTRG.60895.2–mdm-miR393–MD17G1009000 may participate in the anthocyanin metabolism in the fruit. These findings provide a comprehensive view of potential functions for lncRNAs, corresponding target genes, and related lncRNA–miRNA–mRNA networks, which will increase our knowledge of the underlying development mechanism in apple.

## 1. Introduction

Long non-coding RNAs (lncRNAs) are a group of poorly conserved RNA molecules that are longer than 200 nucleotides and have no protein encoding abilities [1,2]. They interact with large molecules, such as DNA, RNA, and proteins. Other known functions of lncRNAs consists of regulating protein modification, chromatin remodeling, proteins’ functional activity, and RNA metabolism in vivo through cis or trans activation at the transcriptional, post-transcriptional, and epigenetic levels [3]. Most lncRNAs exhibit specific spatial structures and spatiotemporal expression patterns. According to the positions of lncRNAs relative to adjacent protein-coding genes in the genome, lncRNAs can be divided into the following five types: sense lncRNAs, antisense lncRNAs, bidirectional lncRNAs, intronic lncRNAs (incRNAs), and large intergenic lncRNAs (lincRNAs) [4,5,6,7].

In plants, lncRNAs are involved in diverse biological processes such as phosphate homeostasis, flowering, photomorphogenesis, the stress response, and fertility [8,9]. As previously reviewed in greater detail, they also play the following important roles: (1) being processed into shorter ncRNAs [10]; (2) acting as both targets and endogenous target mimics for the miRNAs [11,12,13,14]; (3) repressing histone-modifying activities and directing epigenetic silencing through interaction with specific chromatin domains [15,16,17,18]; (4) functioning as molecular cargo for protein relocalization [19,20]; and (5) regulating post-translational processes via protein modifications and protein–protein interactions [21].

LncRNAs can function during tissue development, during sexual reproduction, and in response to external stimuli such as drought, salinity, heat stress, and infections in plants [22,23,24,25,26]. *NATs*, *HID1*, *APOLO*, *ASCO*, *COLDAIR*, *COOLAIR* (*Arabidopsis*) [14,16,27,28,29], *MtENOD40* (*Medicago truncatula*) [19], *LDMAR* (rice) lncRNAs [30,31], which are associated with diverse biological processes, were identified and their physiological functions were initially demonstrated. Most of the knowledge on lncRNAs’ regulatory networks was derived from animals and model plants. To date, only a few lncRNA mechanisms have been revealed in apples, so little is known about the systematic and consensus lncRNA regulatory networks. Further research into apple lncRNAs is warranted to elucidate the regulatory networks of the lncRNAs in apples.

In the present study, we identified lncRNAs from five tissues of ‘Gala’ apples. A total of 9440 lncRNAs were obtained, some of which showed tissue-specific gene expression. The lncRNA–mRNA and lncRNA–miRNA–mRNA interaction-based functional roles showed that these lncRNAs mediated the regulation of photosynthesis-antenna proteins, single-organism metabolic process, glutathione metabolism, and flower coloration. These findings enhance our understanding of the putative regulatory functions of lncRNAs in apple development.

## 2. Results

### 2.1. Identification and Characteristics of lncRNAs in ‘Gala’ Apples

In total, 318.52 Gb of clean data were produced for the RNA-seq from roots, phloem, leaves, flowers, and fruit of ‘Gala’ apples. After mapping them to the apple reference genome, 147,848,711, 142,596,329, 145,428,347, 127,673,515, and 146,376,167 raw reads corresponding to 119,295,520, 116,145,301, 99,284,229, 104,031,259, and 117,814,110 valid reads were identified in the roots, phloem, leaves, flowers, and fruit from the RNA sequencing (Appendix A). On average, the ratio of valid reads, Q30, and GC content in the five libraries were 78.4%, 94.5% and 46.0%, respectively (Appendix A).

Not all reads obtained by RNA-seq were lncRNAs, and transcripts longer than 200 nt with at least two exons were selected as lncRNA candidates. Further screened using coding potential calculator (cpc)/coding-non-coding index (cnci)/protein family (pfam)/coding potential assessment tool (cpat) was conducted, which differentiated protein-coding genes from non-coding genes (Figure 1a). A total of 9440 lncRNAs were obtained from the five tissues of ‘Gala’ apple. Intergenic lncRNAs, sense lncRNAs, antisense lncRNAs and intronic lncRNA accounted for 73.9%, 15.1%, 7.7%, and 3.3% of them, respectively (Appendix A and Figure 1b). The lengths of the lncRNAs ranged from 202 to 10,450 bp. The majority (49.53%) of lncRNA’s lengths were around 400–600 bp (Appendix A and Figure 1b).

The Circos plot revealed a non-random distribution of lncRNAs in the chromosomes. Some chromosomal regions had few lncRNAs, and some had a high density of lncRNAs. The lncRNAs were more widely distributed in chromosomes 5, 10, and 15, which accounted for 7.56%, 7.07%, and 6.85%, respectively, while chromosome 14 had the fewest, at 291. Different types of lncRNAs on chromosomes had differences in their distribution characteristics. Sense lncRNAs were mainly distributed at both ends of chromosomes. They had the highest number of lincRNAs and the distribution was dense. Intronic lncRNAs were sparsely distributed on chromosomes, but the distribution density varied greatly in different regions. Antisense lncRNAs generally tended to be located at one end of the chromosome (Appendix A and Figure 1c).

### 2.2. Comparative Analysis of mRNAs and lncRNAs

We were able to discover the differences in the structure and sequence of mRNAs and lncRNAs by comparing the lengths, numbers of exons and ORFs. The sequence lengths of the lncRNAs were below 1000 bp, accounting for 70.19% of the total, with most being below 400 bp, while the sequence lengths of the mRNAs were mainly over 3000 bp, accounting for 25.74% (Figure 2a,b). Regarding the number of exons in each lncRNA, most of the lncRNAs had two exons, which accounted for 71.99% of the total; the greatest number of exons in one lncRNA was 15. However, slightly over 5% of mRNAs had only one exon, and a single mRNA could contain more than thirty exons (Figure 2c,d). The ORF lengths of the lncRNAs and mRNAs were predicted, and were concentrated at less than 300 bp. However, the maximum length of the lncRNAs was about 500 bp, while the maximum length of the mRNAs was more than 2000 bp, which was related to their respective average lengths; the lncRNAs themselves were relatively short (Figure 2e,f). By comparing the expression levels of the mRNAs and lncRNAs using an FPKM boxplot of all the transcripts, the expression levels of the two were found to be similar, with low expression levels in most of them and high expression levels in some. On the median line, the mRNA expression was slightly lower than the lncRNA expression (Figure 2g, Appendix A).

Alternative splicing is a very common phenomenon in eukaryotic plants. The precursor mRNA (pre-mRNA) of a single gene can be processed at different splicing sites to produce multiple mature mRNA subtypes, which is called alternative splicing. About 60% of multi-exon genes can undergo alternative splicing in plants, which plays an important role in plant growth and development. The number of alternative-splicing isomers of the lncRNAs was significantly lower than that of the mRNAs when examining the screened lncRNAs. Overall, 1336 lncRNAs had more than two isomers, with the most being thirty five isomers for one of the lncRNAs; 26,400 mRNAs had more than two isomers. The highest number of isomers was found for an mRNA with 302 isomers. This indicated that mRNAs had more variable splicing than lncRNAs (Figure 2h).

### 2.3. Differential Expression of lncRNAs in Tissues of ‘Gala’ Apples

To evaluate the prevalence and spatial expression of lncRNAs in ‘Gala’ apples, we analyzed the expression of lncRNAs in roots, phloem, leaves, flowers, and fruit. The prevalence was relatively low in different tissues. In total, 2676 lncRNAs were shared in the five libraries, while 346, 491, 243, 721, and 184 lncRNAs were expressed only in the roots, phloem, leaves, flowers, and fruit (Appendix A and Figure 3a). We normalized the expression profiles of the 2676 prevalent lncRNAs based on their FPKM values, which permitted us to conduct quantitative comparisons of the levels of lncRNAs among the different tissues. Significant differences among the five tissues were found except between the flowers and leaves; the expression of lncRNAs in the fruit was lower than that in the other tissues (Figure 3b). Interestingly, MSTRG.120812.1, NSTRG.120812.2, MSTRG.120812.3, MSTRG.65728.1, and MSTRG.51285.3 were enriched in all five tissues, the first three of which came from Chr15, and the last two of which came from Chr08 and Chr06. This indicated that these lncRNAs had the important basic functions for different tissues in apples.

### 2.4. Analysis of miRNA Precursors in lncRNAs

The lncRNAs that were precursors of miRNAs were predicted by comparing them with miRNA sequences in the miRBase database. We identified 88 lncRNAs as potential precursors for 74 miRNAs belonging to 18 families. One lncRNA could act as a precursor for several miRNAs. For example, MSTRG.121641.28 could be a precursor for five members of the mdm-miR156 families. Furthermore, the same miRNA could have more than one precursor. For example, there were 10 lncRNAs that could be precursors for mdm-miR156a (Appendix A).

### 2.5. Target Gene Prediction for lncRNAs

We used two prediction methods based on the interaction modes for lncRNAs and target genes. First, lncRNAs putatively regulate the expression of their neighboring genes, which can be cis-target genes if they are within the 100 kb of the lncRNAs. There were 9410 lncRNAs that were predicted to have one or more target genes, and there were even 44 target genes for one lncRNA (Appendix A). Second, in the LncTAR software, complementary sequences were used between lncRNAs and mRNAs to calculate the pairing site free energy and standardized free energy, and those below the standardized free-energy threshold (<−0.1) were considered to be the trans-target genes of the lncRNAs. There were only 1218 lncRNAs that were predicted to have target genes, many of which only had one target gene, but MSTRG.41634.1 had 39 trans-target genes (Appendix A).

### 2.6. Analysis of Function of lncRNAs

LncRNAs may serve as precursors of miRNAs and also have cis-targeting and trans-targeting relationships with genes. Therefore, it was speculated that lncRNAs can not only play a role in the regulation network for miRNAs but also directly regulate the expression of target genes. In order to understand the possible functions of lncRNAs, we analyzed the tissue-specific networks for miRNA binding and the functions of cis-target and trans-target genes.

Among the 88 lncRNAs that were predicted to be the precursors of miRNAs, 21 were expressed in all five tissues. The results indicated that their functions were relatively common and important. These mainly served as precursors for mdm-miR156, mdm-miR160, mdm-miR164, mdm-miR166, mdm-miR167, mdm-miR171, mdm-miR172, mdm-miR482, mdm-miR535, mdm-miR828, and mdm-miR5225. Two lncRNAs, MSTRG.17430.4 and MSTRG.10845.2, were specifically expressed in the flowers. These two lncRNAs were predicted to be precursors of mdm-miR156 and mdm-miR160, respectively. Six lncRNAs were specifically expressed in the leaves, which were predicted to be precursors of mdm-miR164, mdm-miR167, mdm-miR172, and mdm-miR5225. LncRNA MSTRG.63448.2 was specifically expressed in the roots, which was predicted to be a precursor of mdm-miR156. LncRNA MSTRG.60895.2 was specifically expressed in the fruit, which was predicted to be a precursor of mdm-miR393 and mdm-miR7126.

The GO term enrichment analysis of the differentially expressed cis-target genes and trans-target genes of the lncRNAs in the five tissues allowed us to understand the biological functions of the differentially expressed genes. For the cis-target genes, 30 main functions were selected for analysis, among which the intracellular part was the most enriched, with 1889 genes, but there was no significant difference among the tissues. The expression levels of genes in the GO-term enrichment of single-organism signaling, ribonucleoprotein complex binding, and extracellular organelles showed large differences among the five tissues. For example, the flowers and fruit had a lower expression of single organism signaling, while the phloem had a higher expression level (Figure 4a). The KEGG pathway enrichment analysis showed that there were significant differences in different tissues for photosynthesis-antenna proteins, with the highest expression in the leaves and the lowest in the root. In addition, the phenylalanine metabolism was the highest in the fruit, but lower in other tissues (Figure 4b). For the trans-target genes, there were only three GO-term enrichment clusters, which were as follows: ion binding, single-organism metabolic process, and heterocyclic compound binding. Roots and fruit demonstrated a higher expression of these than the flowers, phloem, and leaves (Figure 4c). Three KEGG pathway enrichments were formed for trans-target genes, including glutathione metabolism, phagosome, and glycosylphosphatidylinositol (GPI)-anchor biosynthesis; the flowers had a higher expression for glutathione metabolism than the others (Figure 4d).

### 2.7. lncRNA-miRNA Interaction Network in Apples

In order to understand whether lncRNAs could be targets of miRNAs and further affect the post-transcriptional regulation of genes in apples, the potential regulation networks between lncRNAs and miRNAs were predicted. Only the known apple miRNAs and expectation scores ≤ 3 were analyzed here. In total, 1341 possible interrelations between 187 mdm-miRNAs and 174 lncRNAs (1.84%) were predicted. Multiple lncRNAs (or mdm-miRNAs) were predicted to interact with at least one mdm-miRNA (or lncRNA). For example, MSTRG.24932.13 could be targeted by 32 mdm-miRNAs, and mdm-miR172a could target 26 lncRNAs. The mdm-miR156 family had the most target sites (408) in lncRNAs, followed by the mdm-miR172 family with (370). It indicated that the networks between lncRNAs and miRNAs played a vitally important role in apples, especially for the mdm-miR156 and mdm-miR172 families (Appendix A).

Furthermore, miRNAs also had the characteristics of spatial–temporal expression; interactions between lncRNAs and miRNAs were featured to illustrate the functions in different tissues. There were 291 interactions in all five tissues between 26 miRNA families and 55 lncRNAs, including mdm-miR156, mdm-miR164, mdm-miR172, and mdm-miR482. The greatest number of interactions was identified for the mdm-miR156 family, and there were 81 interactions with lncRNAs out of 291 interactions. There were only 71, 1, 111, 41, and 12 interactions in the roots, phloem, leaves, flowers, and fruit, respectively (Figure 5a and Appendix A). In particular, the module in a network of MSTRG.121644.5–mdm-miR156, MSTRG.121644.8–mdm-miR156, MSTRG.2929.2–mdm-miR827, MSTRG.3953.2–mdm-miR395, MSTRG.63448.2–mdm-miR156, MSTRG.9870.2–mdm-miR159, and MSTRG.9870.3–mdm-miR159 were only predicted in the roots. MSTRG.11457.2–mdm-miR7126, MSTRG.138614.2–mdm-miR10993, and MSTRG.60895.2–mdm-miR393 were only predicted in the fruit. They may have special expression and regulation characteristics in these tissues.

In order to further reveal the potential functions of lncRNAs, lncRNA–miRNA–mRNA networks were constructed. We analyzed the correlation of the expression of lncRNAs and mRNAs in five tissues. The expression of lncRNAs and mRNAs targeted by miRNAs should be positively correlated. We extracted lncRNAs and mRNAs whose correlation coefficients were greater than 0.8 with a significance level of 0.05 (Appendix A); then, regulatory network diagrams were constructed (Figure 5b and Appendix A) and the possible role of the lncRNA was speculated on through the functional annotation of mRNAs. As Figure 5c showed, the expression of four lncRNAs and seven mRNAs was highly positively correlated in five tissues. They were targeted by the mdm-miR482 family, which participated in the anti-stress response in apples. Thus, MSTRG.76051.5, MSTRG.103191.1, MSTRG.103191.2, and MSTRG.103191.4 may play a major role in resistance for apples. In addition, other lncRNAs had special roles in different tissues of the apple. For example, MSTRG.60895.2 may participate in the anthocyanin metabolism in fruit by competing with MD17G1009000 as a target of mdm-miR393, whose auxin signaling F-box 2 was associated with anthocyanin metabolism (Appendix A).

### 2.8. qRT-PCR Validation

To confirm our identification of lncRNAs, four random lncRNA candidates were selected for experimental validation using a quantitative reverse transcription PCR (qRT-PCR). The primers were designed based on sequences on both sides of the candidate lncRNAs. As shown in Figure 6, in this study, four randomly selected lncRNAs all showed expression patterns consistent with the RNA-seq results. The R squared values for the RNA-seq vs. qRT-PCR were 0.6137, 0.8999, 0.8593 and 0.9567 for MSTRG.121641.28, MSTRG.121641.3, MSTRG.121644.1 and MSTRG.10845.1, respectively. The results indicated that the lncRNAs identified in the RNA-seq datasets were reliable.

## 3. Discussion

LncRNAs show organ, developmental and environmental expression specificity in plants [32,33,34]. Thousands of plant lncRNAs have been identified, and some of their molecular functions have been studied [18,22,29,35,36,37,38], but only a few studies have been conducted for apples [7,39]. Here, we reported a genome-wide identification and potential function analysis of the lncRNAs in the roots, phloem, leaves, flowers, and fruit of ‘Gala’ apples. Compared with the mRNAs, the lengths of most of the lncRNAs were below 1000 bp, shorter than the mRNAs, and the lncRNAs had fewer exons than the mRNAs, which was consistent with other plants [40,41]. The alternative splicing of mRNAs was more common than that of lncRNAs, the main reason for which may be that the number of exons in the mRNAs was greater than the numbers in lncRNAs.

Studies have indicated that lncRNAs exhibit tissue–specific expression patterns in plants [34,42]. Here, we compared the expression of lncRNAs in five tissues of ‘Gala’ apple; the expression of lncRNAs showed tissue–specific expression. The numbers of lncRNAs identified in different tissues were different, with the highest in the flowers and the lowest in the fruit, which was consistent with the differential functions and organizations of the tissues. Even the co-expressed lncRNAs in the five tissues showed significant differences in their expression levels, indicating that the same lncRNAs had different functions in different tissues, or their contributions to the function were different. It was interesting that the expression levels in the leaves and flowers had no significant differences. This was because the expression levels of several lncRNA were higher in leaves and in flowers, but were lower in other three tissues. In particular, the FPKM of MSTRG.51285.3 in leaves and flowers was 2708.85 and 3666.36, but it was 73.56, 297.62, and 28.80 in phloem, roots, and fruit, respectively. The FPKM of MSTRG.51285.3 accounted for 4.16%, 0.14%, 4.72%, 0.44%, and 0.22% of the total FPKM of 2676 lncRNAs in flowers, phloem, leaves, roots, and fruits, respectively. Unfortunately, we did not obtain its function annotation, either on the regulation of genes or on the interaction with miRNAs. It may work in other ways, and it is worth further exploring its functionality.

An lncRNA mainly regulates its target genes in two ways. One is the cis-regulation of neighboring genes, and the other is the trans-regulation of genes with complementary sequences [43,44]. *DROUGHT INDUCED* lncRNA (DRIR) enhanced drought and salt stress tolerance in *Arabidopsis thaliana* by regulating the expression of genes involved in abscisic acid (ABA) signaling, water transport, and stress relief processes [45]. Obviously, in this study, there were more cis-target genes than trans-target genes. The tissue specificity of the cis-target genes was consistent with that of lncRNAs, since the positively regulated genes were near lncRNAs. However, the target genes subjected to trans–regulation were different, and the trans-target genes were predicted based on complementary sequences, which could better highlight the organization orientation and function. The leaves had the highest level of expression for photosynthesis-antenna proteins, which were located on the thylakoid membranes in the chloroplast, and captured light energy which was transferred to the photosynthetic reaction center [46], although there were low levels in the roots. This aligned with the photosynthetic function of leaves. This suggested that lncRNAs may play an important role in photosynthesis. Two lncRNAs, *COOLAIR* and *COLDAIR*, were found to be present in the locus of *FLC* genes and to critically regulate the *FLC* genes’ expression by transcriptional regulation and histone modification, respectively [16,27]. The flowers had the highest expression for glutathione metabolism. Previous research showed that flowering was promoted by increasing endogenous glutathione [47]. Hence, lncRNAs may interact with glutathione metabolism to improve flowering in apples.

LncRNAs can act as precursors for miRNAs and play a regulatory role through miRNA production. Both miRNAs and lncRNAs have tissue-specific functions. The genome-scale RNA-seq analysis of flower and fruit tissues from *Fragaria vesca* proved that lncRNAs exhibited tissue-specific expression [48]. Of the 2676 prevalent lncRNAs, 21 can serve as precursors for miRNAs, mainly including the mdm-miR156, mdm-miR160, mdm-miR164, mdm-miR166, mdm-miR167, mdm-miR482 families, and other families, indicating that the role of these 21 lncRNAs as miRNA precursors was universal. Regarding tissue-specific lncRNAs, MSTRG.17430.4 and MSTRG.10845.2 were specifically expressed in flowers and predicted to be precursors of mdm-miR156 and mdm-miR160, respectively. MiR156 plays an important role in the flowering of plants [49]. LincRNAs can act as precursors of small RNAs and regulate diverse metabolic pathways. They can alter miRNA expression by interacting with the corresponding miRNAs [50]. It was speculated that MSTRG.17430.4 had a regulatory effect on flowering. Mdm-miR160 could regulate anthocyanin metabolism. The overexpression of mdm-miR160 reduced the content of anthocyanin in flowers [51]. Consequently, MSTRG.10845.2 may play an important role in flower coloration. The present findings suggested that miR156 regulated root development, nitrogen-fixation activity, and root biomass levels [52,53,54]. MSTRG.63448.2 was predicted to be a precursor of mdm-miR156 in the roots, and miR156 regulates a network of downstream genes to affect the growth and development of roots.

LncRNAs and miRNAs interact with each other, imposing an additional level of post-transcriptional regulation. LncRNAs can compete with mRNAs for miRNA molecules, promoting the regulation of miRNA-mediated target repression. This type of ceRNA crosstalk has been widely observed in different biological processes and diseases [12,55,56,57]. In this study, 1341 possible interrelations between 187 mdm-miRNAs and 174 lncRNAs (1.84%) were predicted. The sequences of the binding sites of the lncRNAs were found to be conserved. Thus, a single lncRNA can effectively bind to multiple miRNAs [58]. Multiple lncRNAs (or mdm-miRNAs) were predicted to interact with more than one mdm-miRNA (or lncRNA). *INDUCED BY PHOSPHATE STARVATION 1* (*IPS1*) was a classic example of a functionally characterized cytoplasmic lncRNA that acts via target mimicry to sequester miR-399, which canonically targeted *PHOSPHATE2* (*PHO2*) mRNA [12]. Our study showed similar results, whereby the mdm-miR156 family had the most interactions with lncRNAs in all five tissues. MiR156 played an important role in plant development, anabolism and abiotic stress, and was a regulatory hub for plant growth and development [59,60]. Therefore, specific lncRNAs may play a major role in the apple’s different tissues during its life through ceRNAs for the mdm-miR156 family. It was worth noting that there were more interactions in the roots between mdm-miR156 and lncRNAs than those in fruit. MiR156 involved biotic and abiotic stress in the roots [61,62,63] and fruit ripening [64,65], whereas lncRNAs–mdm-miR156 networks played a broader role in the roots than in the fruit in apples.

LncRNAs and miRNAs reveal spatiotemporal expression specificity [41,66], with specific networks playing a particular role in organization. There were some interactions predicted only in specific tissues. For example, interactions between some lncRNAs and the mdm-miR156, mdm-miR159, mdm-miR395, mdm-miR827 families were only predicted in the roots. Regarding the functionalities of these families, mdm-miR156 was identified as a positive regulator of drought resistance in apples [67]. MiR159 was identified as a post-transcriptional repressor of primary root growth in *Arabidopsis thaliana* [68]. Mi395 was identified as a general component of the sulfate assimilation regulatory network in *Arabidopsis* [69]. MiR827 could enhance drought tolerance in transgenic barley [70]. All of these involved functions in roots; therefore, we concluded that MSTRG.121644.5, MSTRG.121644.8, MSTRG.2929.2, MSTRG.3953.2, MSTRG.63448.2, MSTRG.9870.2, and MSTRG.9870.3, which were only expressed in roots and interacted with miRNAs, could participate in the functions of roots as ceRNAs. In the same way, mdm-miR393, mdm-miR7126, and mdm-miR10993 interacted with miRNAs only in the fruit, although miR393 could regulate fruit/seed set development in cucumbers [71]. *MdARF3* (MD17G1009000) was reported to negatively regulate anthocyanin anabolism as a fast auxin response factor [72]. We had reason to speculate that MSTRG.60895.2–mdm-miR393–MD17G1009000 may participate in the anthocyanin metabolism in the fruit of apples.

## 4. Materials and Methods

### 4.1. Plant Materials

The materials of the phloem, leaves, flowers, and fruit were collected from the ‘Gala’ apples. The scion of ‘Gala’ was grafted on *Malus*
*baccata* (L.) Borkh. in 2001, and the tree was planted in the field of the National Repository of Apple Germplasm Resources (Xingcheng) in 2002. The roots were harvested from the tissue culture plantlets. All the materials were frozen using liquid nitrogen and stored at −80 °C. Each line had three biological replicates.

### 4.2. RNA Isolation, Quantification and Qualification

Total RNA was isolated from each sample using the plant RNA isolation kit (Aidlab Company, Beijing, China) according to the manufacturer’s instructions. rRNA was removed using the Epicenter Ribo-Zero^TM^ (Epicentre, Madison, WI, USA) following the manufacturer’s procedure. RNA degradation and contamination, especially DNA contamination, were monitored on 1.5% agarose gels. The RNA concentration and purity were measured by using the NanoDrop 2000 Spectrophotometer (Thermo Fisher Scientific, Wilmington, DE, USA). The RNA integrity was assessed by using the RNA Nano 6000 Assay Kit of the Agilent Bioanalyzer 2100 System (Agilent Technologies, Santa Clara, CA, USA).

### 4.3. Library Preparation for lncRNA-Seq

A total of 1.5 μg of RNA per sample was used as input material for rRNA removal by using the Ribo-Zero rRNA Removal Kit (Epicentre, Madison, WI, USA). Sequencing libraries were generated by using the NEBNext^R^ Ultra^TM^ Directional RNA Library Prep Kit for Illumina^R^ (NEB, Ipswich, MA, USA) following the manufacturer’s recommendations and index codes were added to attribute sequences to each sample. Briefly, fragmentation was carried out using divalent cations under high temperature in NEBNext First Strand Synthesis Reaction Buffer (5x). First-strand cDNA was synthesized using random hexamer primers and reverse transcriptase. Second-strand cDNA synthesis was subsequently performed using DNA Polymerase I and RNase H. Remaining overhangs were converted into blunt ends via exonuclease/polymerase activities. After the adenylation of the 3′ ends of the DNA fragments, the NEBNext Adaptor with a hairpin loop structure was ligated to prepare for hybridization. In order to select insert fragments, preferentially of 150~200 bp in length, the library fragments were purified with AMPure XP Beads (Beckman Coulter, Beverly, MA, USA). Then, 3 μL of USER Enzyme (NEB, Ipswich, MA, USA) was used with size-selected, and adapter-ligated cDNA at 37 °C for 15 min before PCR. Then, PCR was performed with Phusion High-Fidelity DNA polymerase, Universal PCR primers and an Index(X) Primer. Finally, the PCR products were purified byAMPure XP system (Beckman Coulter, Beverly, MA, USA) and the library quality was assessed on the Agilent Bioanalyzer 2100 (Agilent Technologies, Santa Clara, CA, USA) and by qPCR.

### 4.4. Clustering and Sequencing

The index-coded samples were clustered using the acBot Cluster Generation System with the TruSeq PE Cluster Kitv3-cBot-HS (Illumina, NEB, Ipswich, MA, USA) according to the manufacturer’s instructions. After cluster generation, the library preparations were sequenced on an Illumina Hiseq platform, and paired-end reads were generated.

### 4.5. Quality Control

The raw data (raw reads) of FASTQ format were first processed through in-house Perl scripts. In this step, the clean data (clean reads) were obtained by removing adapters from reads, reads containing ploy-N, and low quality reads from the raw data. At the same time, the Q30, GC content, and sequence duplication level of the clean data were calculated. All the downstream analyses were based on clean data with high quality.

### 4.6. LncRNA Analysis

The transcriptome was assembled using StringTie based on the reads mapped to the reference genome (*Malus* × *domestica* GDDH13 v1.1) (www.rosaceae.org, accessed on 16 May 2022). The assembled transcripts were annotated using the GffCompare program. The unknown transcripts were used to screen for putative lncRNAs. Four computational approaches including cpc/cnci/pfam/cpat were combined to sort non-protein coding RNA candidates from putative protein-coding RNAs in the unknown transcripts. Putative protein-coding RNAs were removed using a minimum length and exon number threshold. Transcripts longer than 200 nt and with at least two exons were selected as lncRNA candidates and further screened using cpc/cnci/pfam/cpat, which has the power to distinguish protein-coding genes from non-coding genes. Furthermore, different types of lncRNAs including lincRNAs, intronic lncRNAs, antisense lncRNAs, and sense lncRNAs were selected using Cuffcompare (http://cole-trapnell-lab.github.io/cufflinks/cuffcompare/index.html, accessed on 16 May 2022).

### 4.7. Quantification of Gene Expression Levels

StringTie (1.3.1) (https://ccb.jhu.edu/software/stringtie/index.shtml, accessed on 16 May 2022) was used to calculate the FPKMs of both the lncRNAs and coding genes in each sample [73]. The gene FPKMs were computed by summing the FPKMs of the transcripts in each gene group. The FPKM, which means fragments per kilo-base of exon per million fragments mapped, was calculated based on the lengths of the fragments and the reads count mapped to these fragment.

### 4.8. Differential Expression Analysis

The differential expression analysis of two conditions/groups was performed using the DESeq R package (1.10.1) (http://www.bioconductor.org/packages/release/bioc/html/DESeq.html, accessed on 16 May 2022). DESeq provides statistical routines for determining differential expression in digital gene expression data using a model based on the negative binomial distribution. The resulting *p*-values were adjusted using the Benjamini–Hochberg approach for controlling the false-discovery rate. Genes with adjusted *p*-values < 0.01 and absolute values of log2 (Fold change) > 1 as determined by DESeq were assigned as differentially expressed.

### 4.9. Gene Functional Annotation

Gene functions were annotated based on the following databases:Nr (NCBI non-redundant protein sequences).Pfam (Protein family).KOG/COG (Clusters of Orthologous Groups of proteins).Swiss-Prot (a manually annotated and reviewed protein sequence database).KEGG (Kyoto Encyclopedia of Genes and Genomes).GO (Gene Ontology).GO enrichment analysis of the differentially expressed genes (DEGs) was implemented using the topGO R packages.

We used the KOBAS [74] software to test the statistical enrichment of differentially expressed genes in KEGG pathways.

The sequences of the DEGs were blast (blastx) to the genome of a related species (the protein–protein interactions which exist in the STRING database: http://string-db.org/, accessed on 16 May 2022) to obtain the predicted PPI of these DEGs. Then, the PPIs of these DEGs were visualized in Cytoscape [75].

### 4.10. Prediction of miRNA Target Sites in lncRNAs

We identified the targets of miRNAs using TargetFinder, based on the known miRNAs of apples, the newly predicted miRNAs, and the gene sequence information for ‘Golden delicious’. Interactions between lncRNAs and miRNAs with expectation score ≤ 3 were selected. As lncRNAs contain multiple miRNA binding sites, the miRNA target gene prediction methods can be used to identify the lncRNAs that bind to miRNAs, and the functions of lncRNAs can be elucidated based on the functional annotation of the miRNA target genes.

### 4.11. Quantitative Real-Time PCR Validation

Quantitative real-time PCR (qRT-PCR) was carried out to validate the levels of differential expressional lncRNAs from ‘Gala’ apples. According to the instructions of the TruScript First-strand cDNA SYNTHESIS Kit (Aidlab Company, Beijing, China), 800 ng of the total RNA was reverse-transcribed with random primers. The reaction system included 800 ng of RNA, 4 μL of 5 × RT Reaction Mix, 0.5 μL of rondam primers/oligodT, 0.5 μL N6, 0.8 μL of TruScript H^−^ RTase/RI Mix, and RNase free dH_2_O was added to obtain a 20 μL volume; the reaction conditions were 42 °C for 40 min and 65 °C for 10 min.

Primers were designed in order to obtain the amplicon from the template (Appendix A). Quantitative real-time PCR (qRT-PCR) was performed using SYBR Green Master Mix (Vazyme, Nanjing, China). The qRT-PCR aliquot contained 1 μL of cDNA, 3 μL of ddH_2_O, 0.5 μL of each of the forward and reverse primers (200 nM), and 5 μL of 2 × SYBR^®^ Green Supermix. The reaction conditions included initial denaturation at 95 °C for 3 min, followed by 39 cycles at 95 °C for 10 s, and 60 °C for 30 s, with melt curve analysis (60~95 °C, +1 °C/cycle; holding time: 4 s). The levels of the lncRNAs were normalized to *q*Actin. All the real-time PCR assays were performed with three biological replicates. The relative expression levels were calculated with the 2^−ΔΔCt^ method [76].

## 5. Conclusions

We identified lncRNAs in five tissues of the ‘Gala’ apple, a variety widely cultivated worldwide. A total of 9440 unique lncRNAs were identified from the leaves, phloem, flower, fruit, and roots. Cis-target and trans-target genes prediction for lncRNAs showed that the target genes were significantly enriched in molecular functions related to photosynthesis-antenna proteins, single-organism metabolic process and glutathione metabolism. In total, 88 lncRNAs were predicted to be precursors of miRNAs. A total of 1341 possible interrelations between 187 mdm-miRNAs and 174 lncRNAs (1.84%) were predicted when performing a search across the miRNAs of *Malus* in miRBase. It was predicted that MSTRG.121644.5, MSTRG.121644.8, MSTRG.2929.2, MSTRG.3953.2, MSTRG.63448.2, MSTRG.9870.2 and MSTRG.9870.3 could participate in the functions of roots as ceRNAs. MSTRG.11457.2, MSTRG.138614.2 and MSTRG.60895.2 could adopt special functions in the fruit by working with miRNAs. Potential lncRNA–miRNA–mRNA networks were constructed, and the possible roles of lncRNAs in different tissues were considered. We had reason to surmise that MSTRG.60895.2 might participate in anthocyanin metabolism in the fruit by competing with MD17G1009000 as a target for mdm-miR393.

## Figures and Tables

**Figure 1 ijms-23-05931-f001:**
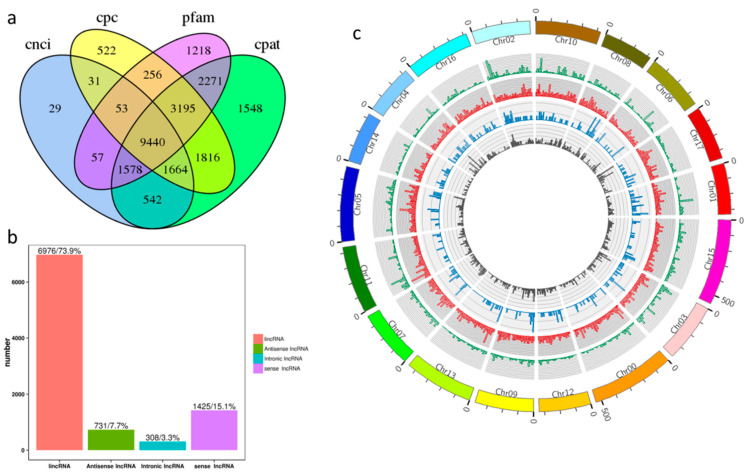
Screening and classification of lncRNAs. (**a**) Results of screening using cnci, cpc, pfam, and cpat. (**b**) Classification of lncRNAs. (**c**) Distribution of lncRNAs identified in chromosomes (green: sense lncRNAs; red: lincRNAs; blue: intronic lncRNAs; grey: antisense lncRNAs).

**Figure 2 ijms-23-05931-f002:**
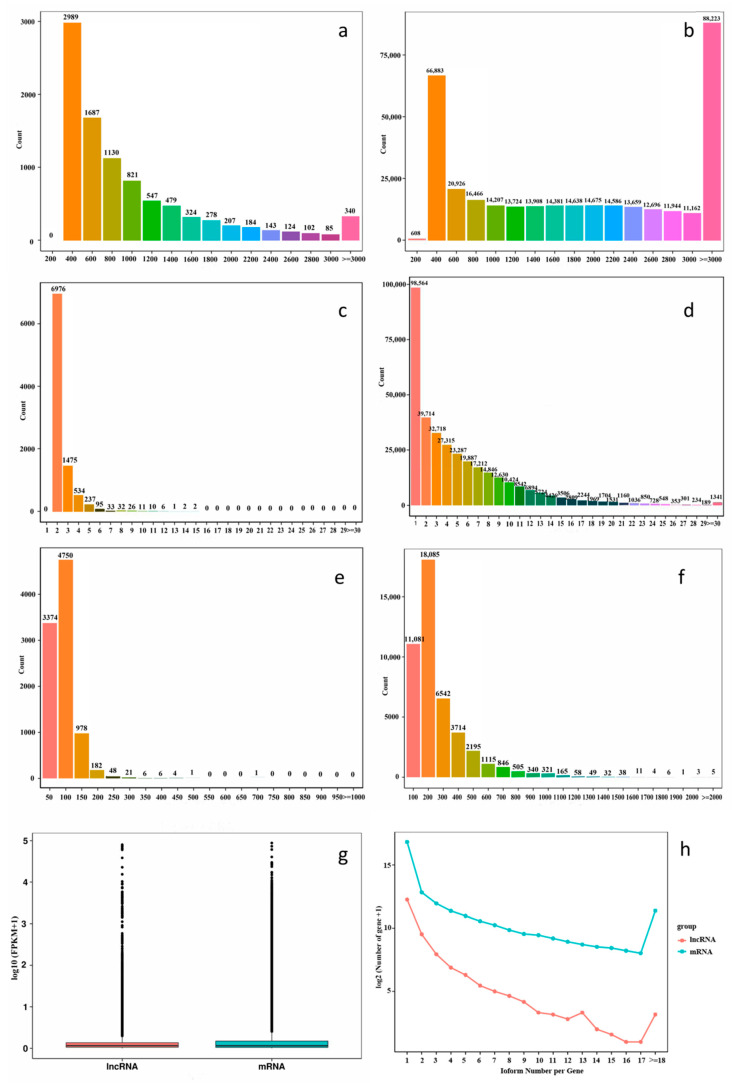
Comparison of the lncRNAs and mRNAs identified. (**a**) Length distribution of the lncRNAs. (**b**) Length distribution of the mRNAs. (**c**) Exon distribution of the lncRNAs. (**d**) Exon distribution of the mRNAs. (**e**) ORF lengths distribution of the lncRNAs. (**f**) ORF lengths distribution of the mRNAs. (**g**) Expression levels of the lncRNAs and mRNAs identified. (**h**) Isoforms of the lncRNAs and mRNAs identified.

**Figure 3 ijms-23-05931-f003:**
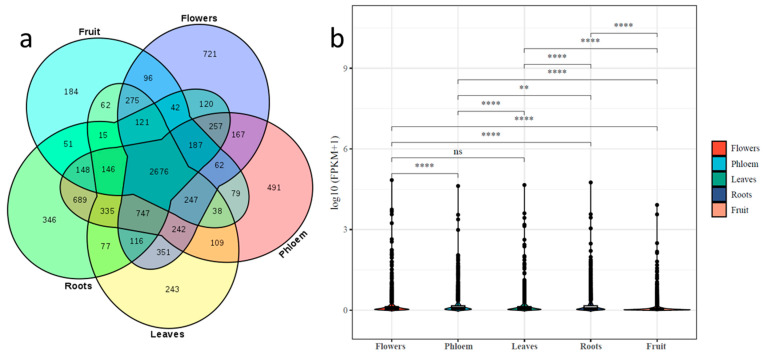
Tissue–specific expression characteristics of the lncRNAs identified. (**a**) Venn diagram showing the number of lncRNAs identified in five tissues of apples. (**b**) The overall abundance patterns of lncRNAs according to FPKM calculations (ns: no significant difference; **: significant difference level at *p* < 0.01; ****: significant difference level at *p* < 0.0001).

**Figure 4 ijms-23-05931-f004:**
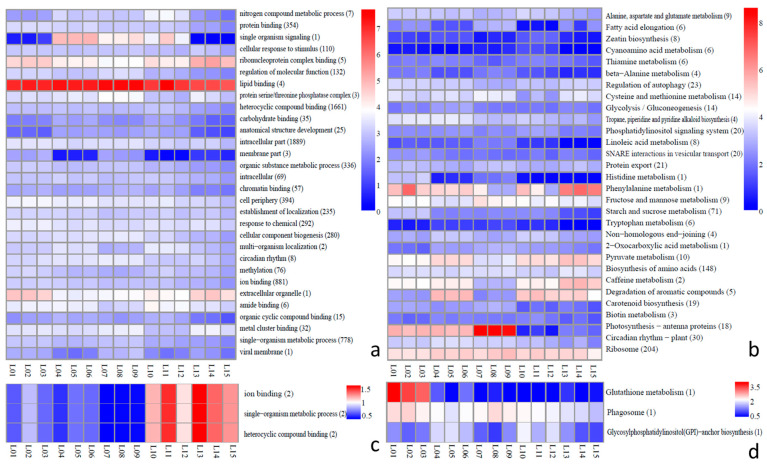
GO term and KEGG pathway enrichment analysis of all differentially expressed of lncRNAs in five tissues (L1–L3: flowers; L4–L6: phloem; L7–L9: leaves; L10–L12: roots; L13–L15: fruit). (**a**) GO term enrichment analysis of all differentially expressed cis-target genes of lncRNAs. (**b**) KEGG pathway enrichment analysis of all differentially expressed cis-target genes of lncRNAs. (**c**) GO term enrichment analysis of all differentially expressed trans-target genes of lncRNAs. (**d**) KEGG pathway enrichment analysis of all differentially expressed trans-target genes of lncRNAs.

**Figure 5 ijms-23-05931-f005:**
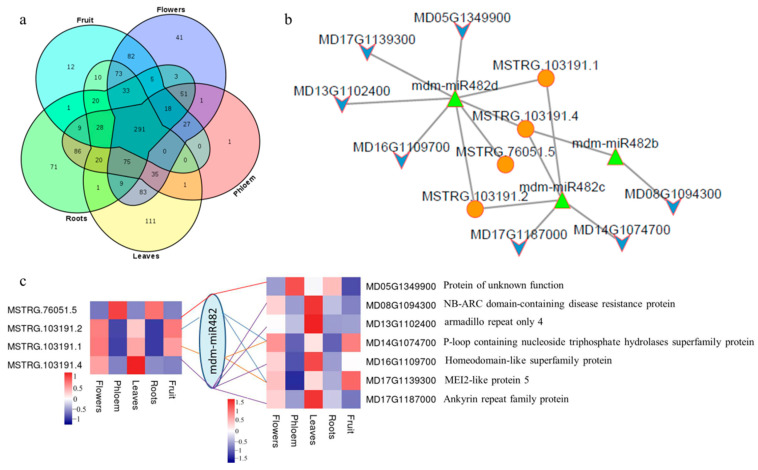
Tissue-specific characteristics and functions, and prediction of lncRNA-miRNA-mRNA. (**a**) Venn diagram of interactions between lncRNAs and miRNAs of five tissues. (**b**) lncRNA-miRNA-mRNA networks for mdm-miR482 (orange circle: lncRNAs; green triangle: mdm-miRNAs; blue arrow: mRNAs). (**c**) Pearson correlation coefficient for expression of lncRNAs and mdm-miR482 in five tissues and functions of mRNAs.

**Figure 6 ijms-23-05931-f006:**
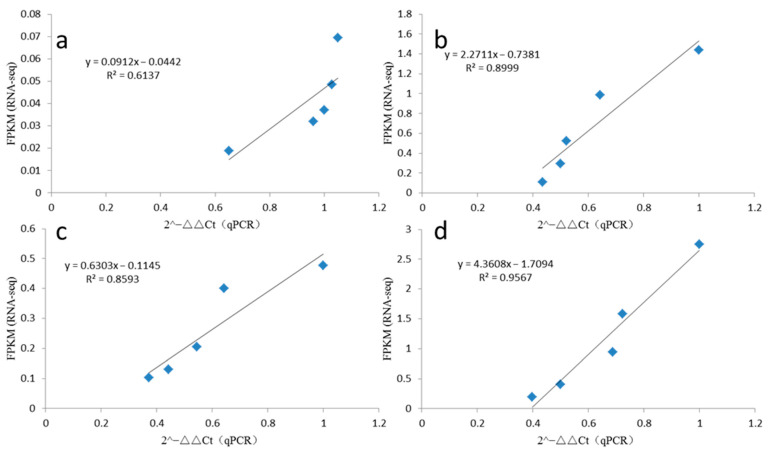
The linear relationship between RNA-seq and qRT-PCR data ((**a**–**d**) were MSTRG.121641.28, MSTRG.121641.3, MSTRG.121644.1, and MSTRG.10845.1, respectively).

## Data Availability

The raw sequence data reported in this paper have been deposited in the Genome Sequence Archive (Genomics, Proteomics and Bioinformatics 2021) in National Genomics Data Center (Nucleic Acids Res 2022), China National Center for Bioinformation/Beijing Institute of Genomics, Chinese Academy of Sciences (GSA: CRA006669) that are publicly accessible at https://ngdc.cncb.ac.cn/gsa (accessed on 15 March 2022).

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
