# Peer review of "Expression Profiles and Characteristics of Apple lncRNAs in Roots, Phloem, Leaves, Flowers, and Fruit"

_ijms, 2022, doi:10.3390/ijms23115931_

Round 1
Reviewer 1 Report
The work "Expression profiles and characteristics of apple lncRNA in roots, phloem, leaves, flowers and fruits" by Wang et al. (ijms-1718838) is a very interesting manuscript that aims at the characterization of the lncRNAs in different tissues of apple. The authors generated a huge amount of data, and several bioinformatics approaches are undertaken to characterize the possible interactions on these lncRNAs with microRNAs and mRNAs. However, there are several issues that prevent me from recommending the publication of the manuscript in its present form.
First of all, English grammar and vocabulary should be thoroughly checked and corrected. There are many wrong expressions, typos, missing particles and lack of concordance between verbs and subjects. Just a few examples:
- Line 10: "lncRNAs possess the crucial effects". The effects are played or displayed, but not possessed.
- Line 90: "was the most,", line 231: "the most interaction".
- Line 162:"if they neighbor". Neighbor is not a verb.
I will stop here. Authors should rewrite the manuscript, and if needed ask for the help of a professional language service, like the one provided by MDPI (https://www.mdpi.com/authors/english).
Concerning the text itself, it is confusing at times, making it difficult to follow authors´ reasoning. Moreover, there are a few scientific misconceptions that make difficult the understanding of the work:
- Lines 12-13: " predicted the intricate regulatory, " Do you mean intricate regulatory networks? This mistake can be found several times in the manuscript.
- Line 75: "more than two exons". Do you mean with at least two exons?
- In section 2.1, it should be clarified that not all reads are lncRNAs.
- Section 2.2 is extremely confusing, it should be rewritten for clarity purposes. Besides, in Line 114 authors talk about expression trends, which is wrong. These are just expression levels, not trends.
- The text about alternative splicing is unnecessary, but in any case it should need a reference. Besides, intron retention is the most common event in plants.
- Line 142: authors are measuring levels of expression at one given point in different tissues, there is no down-regulation or up-regulation, just different expression levels. Line 146:"had the most basic important function". Please correct.
- Line 161: lncRNAs will putatively regulate the expression of neighbouring genes, but it is something that needs to be proven.
- Table S9: no statistical value is given concerning the identification of putative targets.
- Line 190: "GO rich clustering". These are term enrichment analysis. This mistake can be found several times in the text.
- Line 216: "lncRNAs could be target sites by miRNAs". According to the Introduction in this manuscript, is the other way around. Please correct or clarify your statements.
- Line 234: a single interaction by a lncRNA and a miRNA does not make a network, it could be a module or sub-module in a network.
- Lines 275-284: this is not Discussion, is Results, it should be erased or moved to the previous section.
- Line 293: "obvious organ characteristics". Do you mean tissue-specific expression?
- Line 300: "flower bud and leaf bud are homologous organs". Absolutely not, please correct. The rest of the Discussion is also extremely confusing, should be rewritten.
-Line 420: did the authors removed the reads containing adapters, or did they only removed the adapters from the reads?
- Which version of the Malus domestica genome did the authors used as reference for the mapping? Which program was used for this task?
- Line 437: Stringtie should be referenced (Pertea et al.). Moreover, did the authors also used Ballgown or other software for the generation of the read counts?
- Section 2.10: Which statistical values were used for this analysis?
- Keywords: "tissue" and "spatial expression" should be replaced with "tissue-specific expression". Other possible keywords: regulatory networks.
Overall, the manuscript should be rewritten for clarity purposes, some scientific concepts improved and English grammar and vocabulary corrected. I encourage the authors to follow the recommendations given in order to get their valuable results published.
Reviewer 2 Report
The work is a very interesting study on the analysis of lncRNA in various apple tissues and the functions that lnc RNA molecules can perform in individual tissues. The obtained results allowed to deepen the knowledge about the putative regulatory functions of lncRNA in apple development.
The work is a comprehensive approach to research, the experiment is well designed and presented. My remark concerns supplementing the methodology with the RNA isolation process.
